# Microwave Soldering of Low-Resistance Conductive Joints—Technical and Economic Aspects

**DOI:** 10.3390/ma16093311

**Published:** 2023-04-23

**Authors:** Sorin Vasile Savu, Cristian Daniel Ghelsingher, Iulian Stefan, Nicusor-Alin Sîrbu, Daniela Tarniță, Dalia Simion, Ionel Dănuț Savu, Ionela Gabriela Bucșe, Traian Țunescu

**Affiliations:** 1Department of Engineering and Management of Technological Systems, Faculty of Mechanics, University of Craiova, 200585 Craiova, Romania; iulian.stefan@edu.ucv.ro (I.S.); ionel.savu@edu.ucv.ro (I.D.S.); ionela.bucse@edu.ucv.ro (I.G.B.); 2Doctoral School Academician Radu Voinea, Faculty of Mechanics, University of Craiova, 200585 Craiova, Romania; ghelsingher.cristian.j4s@student.ucv.ro (C.D.G.); tunescu.traian.v7x@student.ucv.ro (T.Ț.); 3NRDI for Welding and Material Testing—ISIM Timișoara, 300222 Timișoara, Romania; asirbu@isim.ro; 4Department of Applied Mechanics, Faculty of Mechanics, University of Craiova, 200585 Craiova, Romania; daniela.tarnita@edu.ucv.ro; 5Department of Finances, Banks and Economic Analysis, Faculty of Economics and Business Administration, University of Craiova, 200585 Craiova, Romania; dalia.simion@edu.ucv.ro

**Keywords:** soldering, microwave heating, thermal field, mechanical characteristics

## Abstract

Soldering processes are applied in the fabrication of electronic circuits used in most modern domestic and industrial technologies. This article aims to introduce a new soldering technology based on the microwave joining of copper materials used in electronic applications. The study was focused on microwave technology used as the thermal source for soldering. A simulation model of temperature distributions in copper plates with overall dimensions of 50 × 10 × 0.8 mm was developed in order to determine the necessary microwave power for soldering. For 270 °C simulated on the surface of copper plates, the microwave-injected power was determined to be 598.89 W. An experimental program for 600, 650, 700, and 750 W was set in order to achieve soldering of copper plates in less than 1 min. Soldered copper plates were subject to electrical resistance measurements being obtained with variations up to ±1.5% of the initial electrical resistance of the base materials. The quality of joints has also been analyzed through microscopy after the soldering process. In addition, mechanical properties were determined using a universal testing machine. The results have shown similar behavior of the samples layered with SAC on the one-side and double-side but also a significantly lower force before breaking for one-side-layered samples. An economic analysis was performed and the results obtained have shown that in terms of energy efficiency and total costs for microwave soldering compared with manual soldering, microwave soldering is cost-effective for industrial processing.

## 1. Introduction

Copper plate soldering is a common technique used in various industries, including the electronic, plumbing, and automotive industries. The process involves joining two copper plates using a solder material that melts at a lower temperature than the copper plates. This method creates a strong bond between the copper plates and is essential in various applications that require a durable and reliable joint.

Solder materials used for copper plate soldering can be categorized into two types: lead-based and lead-free solder materials. Lead-based solder materials have been traditionally used in the past but, due to environmental concerns, they are gradually being phased out. Chen et al. reported that lead-free solder materials are becoming more popular because they are environmentally friendly and comply with the regulations set by RoHS (Restriction of Hazardous Substances) [1]. Lead-free solder materials are further classified into three categories: tin-based, silver-based, and copper-based. Palcut et al. reported that tin-based solder materials are the most commonly used lead-free solder material for copper plate soldering. They have a low melting point and are easy to work with, making them an ideal choice for most applications [2]. In addition, Liao et al. studied how electrochemical migration (ECM) can cause short circuits in bonds. They developed and reported advanced methods of inhibiting ECM [3]. Silver-based solder materials are used in applications that require a stronger joint, such as in high-temperature environments. Sandeep et al. reported concerns related to the prohibition of using high-melting-temperature-type solders. According to their studies, it is important to improve the research in the field of solders in order to develop alternatives of high-lead solders for electronic applications [4]. Copper-based solder materials are used in applications that require a high level of electrical conductivity, such as in the electronics industry. The latest research, performed by Asyraf et al. [5], related to the influence of copper-based alloy on the intermetallic compound formation and growth after laser soldering was focused on mechanical characteristics and wettability by varying the copper percentage [6]. The study reported that the Sn-Cu (0.6–0.7 wt.%) solder alloy provides high shear strength.

There are various techniques involved in copper plate soldering, including manual soldering, wave soldering, and reflow soldering. Manual soldering involves heating the solder material with a soldering iron and applying it to the joint between the base materials. Ultrasonic-assisted soldering of dissimilar materials has been reported by Li et al. They successfully obtained Si/Cu joints using Sn-3.5Ag-4Al solder [7]. Wave soldering involves passing the copper plates through a bath of molten solder material. Reflow soldering involves applying solder paste to the joint and heating it using a reflow oven. Copper plate soldering is not without its challenges. One of the main challenges, defined by Chang et al., is ensuring that the joint is strong and durable [8]. This requires proper preparation of the copper plates, including cleaning and pre-tinning. Another challenge is the risk of thermal damage to the copper plates, which can lead to deformation or even the complete failure of the joint. To overcome this, according to Huang et al., it is essential to use the correct soldering temperature and technique [9].

The purpose of this study is to introduce a new soldering technique by using microwave technology as a thermal source for joining copper base materials. Microwave processing has been developed in the last 30 years as a viable technology for wide areas of applications: sintering, welding, soldering, waste treatment, and even for military purposes. Microwave applications in the industry have been developed by Metaxas [10] in his effort to describe how microwaves can be used for food tempering, rubber vulcanization, vacuum drying, or sintering ceramics. However, microwave technology cannot be applied directly to all classes of materials. Metals, in the solid state, reflect microwaves [11,12,13], but indirect heating can be applied if are used ceramic susceptors [14,15,16,17,18]. On the other hand, most ceramic materials are good absorbers of microwaves and, depending on the loss factor coefficient, can be sintered [19] for obtaining powerful magnets for motors and actuators [20], or inductive sensors for different sensing applications [21].

This research aims to develop microwave soldering of copper plates using indirect microwave heating as the thermal source and Sn-Ag-Cu commercial alloy as solder. The study is focused on microwave heating for soldering as well as the mechanical properties of the joints. In addition, microscopy analysis was performed in order to reveal the quality of the joints from the soldering point of view. Finally, an economic analysis was taken into consideration in order to establish the efficiency of the microwave soldering process.

## 2. Materials and Methods

The microwave soldering process was developed using microwave installation, copper plates as base materials, and insulating fire brick as susceptor in order to obtain good absorbance of the microwaves and high rates of conversion of microwaves into heat. Before starting microwave soldering, a simulation of the thermal transient into insulating fire brick and temperature distribution inside copper plates was developed in order to increase the stability of the process and decrease the soldering time.

### 2.1. Simulation of the Thermal Field for Indirect Microwave Soldering

Metals in the solid state are not good absorbers of microwaves. The reflection coefficient is high and the conversion rate of microwaves into heat is low. Therefore, microwave applications should use ceramic susceptors for increasing the conversion rate. On the other hand, microwave heating requires that parameters related to microwave power input and a heating mechanism be well defined, in order to avoid unwanted phenomena such as thermal runaway or microwave plasma. In this respect, a simulation of the thermal field and temperature distribution inside copper plates must be performed in order to establish the level of microwave power that is injected from the microwave generator to the ceramic susceptor. The simulation model was created taking into consideration the constraints and limitations of the microwave heating chamber which is presented in Figure 1.

The model contains 2 blocks from insulating fire brick material having a parallelepiped shape with overall dimensions 35 × 35 × 20 mm, placed on the bottom of the microwave heating chamber. On the top of the blocks are placed the copper plates, each of them having overall dimensions of 50 × 10 × 0.8 mm. The overlap of copper plates was considered to be 10 mm from each base material. Placed on the top of the copper plates is a parallelepiped shape, made from insulating fire brick and having the overall dimensions 35 × 35 × 10 mm with a central hole of 10mm in diameter. This hole was considered in order to ensure the possibility of measuring the temperature using an infrared pyrometer.

In order to establish the temperature distribution and thermal field applied to copper plates, Fourier’s law related to heat conduction in solids is applied. Starting from basic conduction heating equations [22,23], the temperature distribution can be simulated using Equation (1):(1)$$Q=-k\cdot \left(\frac{dT}{dx}i+\frac{dT}{dy}j+\frac{dT}{dz}k\right)$$
where *Q* (W) is the rate of heat flow through the material, *k* is the thermal conductivities of the insulating fire brick: *k*_ifb_ = 0.3 W/mK, copper: *k*_Cu_ = 401 W/mK and SAC305 alloy: *k*_SAC305_ = 55 W/mK. The vectors *i*, *j*, and *k* are unit vectors in the *x*, *y*, and *z* directions, respectively. According to the model presented in Figure 1, for the application of microwave soldering of cooper plates it is important that the temperature distribution is on the *Y* axis. For the planar layer, the rate of heat is given by Equation (2):(2)$$Q=-k\cdot A\cdot \frac{∆T}{L}=-k\cdot A\cdot \frac{\left(T_{h}-T_{c}\right)}{L}$$
where *A* (mm^2^) is the heating transfer area, *T_h_* (°C) is the highest temperature obtained from the conversion of the microwaves into heat, *T_c_* (°C) = 25 °C is the ambient temperature considered for simulation and *L* (mm) = 58 mm represents the total length of the assembly. The simulation was performed considering thermal transient for a boundary time of the 60 s and temperature limit recorded by infrared pyrometer, on the top of copper plates, up to 270 °C. This temperature is considered more than enough to achieve soldering in the microwave field of the copper plates. On the other hand, the time step size for transient thermal analysis is related to element conduction length and material properties. For the simulation, a time step was chosen by taking into consideration Equation (3):(3)$$∆t\;\leq \frac{\delta^{2}}{4\cdot \sigma }$$
where *σ* (mm^2^/s) is thermal diffusivity calculated as ratio between thermal conductivity *k* = 0.230 W/mK at 260 °C and product between mass density ρ = 0.705 g/cm^3^ and specific heat c = 1.05 kJ/kgK [24,25].

Figure 2 presents the simulation of the thermal transient of the model taking into consideration the boundaries presented above [26].

According to the simulation of thermal transient from Figure 2, the temperature distribution, in insulating fire brick, solder alloy, and copper plates, presents a maximum temperature of 731 °C in the center of the firebrick block, which is normal for microwave heating taking into consideration that is a volume heating, and a minimum temperature on the surface of copper plates of 240 °C. The vertical transient of the temperature distribution can be assessed by plotting nodes from the center of the fire brick to the edge. Figure 3a presents the transient of the temperature as a function of time for selected points from the simulation model, and Figure 3b presents the plot of temperature for 7 different locations on the *Y* axis.

In addition, the simulation revealed, in Figure 4, the transient temperature in copper plates for boundary conditions applied to the selected model. 

According to Figure 4, the thermal transient simulation and temperature distribution revealed an equilibrium of temperature along copper plates with values between 232 °C and 240 °C. These similar values ensure that temperature in copper plates is uniformly distributed, and soldering application can be successfully implemented on the entire surface of the base material. The simulations of the thermal transient for 12 different locations on the *X*-axis and temperature distribution along the *X*-axis are presented in Figure 5a,b.

The highest temperature, 240 °C, was obtained for location 6 (38,0,0) which represents, according to the selected model from Figure 4, the joining point between the copper plates.

### 2.2. Indirect Microwave Soldering of Copper Plates

The microwave soldering of copper plates is not characterized by direct microwave heating of copper and solder due to their high reflection coefficient. Therefore, indirect microwave heating was applied by taking into consideration a ceramic susceptor from insulating fire brick material. The simulation of the thermal transient and temperature distribution revealed a temperature of 731 °C in the center of the fire brick block, in order to obtain at least 240 °C as the soldering temperature of copper plates. Starting from this limitation, in terms of temperature development inside fire brick material, the level of microwave power was calculated using Equation (4) [25]:(4)$$P_{MW}=k\cdot V\cdot f^{2}\cdot \varepsilon \cdot \tan\delta \cdot \frac{∆T}{t}$$
where *P_MW_* (W) is the microwave-injected power from the generator, *k* = 0.1008 × 10^9^ is the proportionality constant depending on the geometry and material properties of the fire brick, *V* = 0.000049 m^3^ is the overall volume of fire brick blocks, *f* = 2450 MHz is the frequency of microwave radiation, *ε* = 4.05 F/m is the permittivity of insulating fire brick, tan *δ* = 0.0002–0.0004 at 2450 MHz is the loss tangent of insulating fire brick, Δ*T* = 1004.15 K = 731 °C is the temperature rise in the fire brick material, and *t* = 60 s is the total time required by the soldering application. By solving Equation (4), the minimum level of injected power was determined to be *P_MW_* = 598.89 W. Then, in order to find out to what extent the level of microwave power influences the sustainability of the soldering process and quality of the joints, the microwave injection was increased gradually to 650 W, 700 W, and 750 W. A total of 16 samples were obtained, 8 of 16 samples being subject of microscopy analysis and the other 8 samples were subject to mechanical testing in order to establish the mechanical properties. All 16 samples have also been subjected to electrical resistance measurement, in order to determine if the total resistance remains at low values.

Before starting the soldering process in the microwave field, the samples were prepared using initial copper plates having 100 mm length, 10 mm width, and 0.8 mm thickness. The initial plates were subjected to “two points” electrical resistance measurement using digital multimeter PCI 2064 DMM 7^1/2^ Signametrics produced by Keysight Technologies (Santa Rosa, CA, USA, 2008). The next step after measurements was the cutting of samples into two equal pieces having the dimensions 50 × 10 × 0.8 mm. Then, each sample obtained after the cutting process has also been subjected to electrical resistance measurement. The values obtained are presented in Table 1 (E1…E4—experimental programs for 600 W, 650 W, 700 W, and 750 W; P1…P2—copper plates having solder layer deposited on one sample or on both samples; T and M—samples, soldered in the same conditions, that were subjected of microscopy and mechanical properties analysis).

The quantities from Table 1 are R_i_ (Ω) is the resistance that was measured on copper plates before cutting in samples, W_i_ (mm) is the thickness of initial copper plates, R_iS1_ (Ω) and R_iS2_ (Ω) are the electrical resistance of the copper plates after the cutting process, W_S1+SAC305_ (mm) and W_S2+SAC305_ (mm) are the thickness of copper samples with SAC305 solder deposited.

The samples from Table 1 were prepared for microwave soldering taking into consideration that each pair of samples were subjected to measurements of electrical resistance after soldering, as well as mechanical testing using a universal testing machine. The codes EP1T (samples used for measurement of electrical resistance after soldering and then for mechanical testing) and EP1M (samples used for measurement of electrical resistance after soldering and then for microscopy analysis) were assigned to samples with solder SAC305 deposited on only one sample. Similarly, the codes EP2T (samples used for measurement of electrical resistance after soldering and then for mechanical testing) and EP2M (samples used for measurement of electrical resistance after soldering and then for microscopy analysis) were assigned to samples with solder SAC305 deposited on both samples subjected to microwave soldering. The SAC305 was deposited on all samples, covering 10 mm length of each sample. The total length of samples was 90 mm, in order to respect the conditions used in the simulation of thermal transient and temperature distribution. The sample used in the experimental program is presented in Figure 6. In order to have identical conditions for microwave soldering samples, EP1T and EP1M were soldered together in one process. Similarly, samples EP2T and EP2M were subjected to soldering in the microwave reactor chamber with the same experimental conditions.

The solder SAC305 layers were deposited manually using the digital soldering station ProsKit 48 W/230 V manufactured by ProsKit Industries (New Taipei City, Taiwan, 2016) with a temperature set at 371 °C. The technical characteristics of solder: SAC305-type Kristall 400, solder alloy Sn97.1Ag2.6Cu0.3 with 0.7 mm. For proper deposition of the solder, rosin was used to maintain a good temperature during the deposition process.

The microwave soldering experimental procedure was supported by a microwave heating installation consisting of a MW-Generator Set 6000 W, 2450 MHz continuous wave containing a magnetron head type MH6000S-251BF from Muegge GmbH (Reichelsheim, Germany, 2020). The microwave generator is driven by an MW-Power Source Supply 6000 W, 2450 MHz, 3 × 400 V, continuous wave type MX6000D-154KL, both manufactured by Muegge GmbH (Reichelsheim, Germany, 2020). Figure 7 presents the microwave installations and auxiliary devices used, during the soldering process, to control and record the data for further analysis.

The main auxiliary device is an infrared pyrometer CSmicro 3M manufactured by Optris GmbH (Berlin, Germany, 2020) with a temperature range between 50 °C to 600 °C, with a spectral range of 2.3 μm which is ideal for the temperature measurement of metallic surfaces. The control of the infrared pyrometer was performed using Compact Connect supplied by the manufacturer together with IR pyrometer. In addition, for matching load impedance, a Tristan auto tuner was used up to 6000 W, 2450 MHz from Muegge GmbH (Reichelsheim, Germany, 2008). The tunning process, related to matching load impedance, was supported by Homer Software provided by SK-Team d.o.o. (Bratislava, Slovakia, 2008).

The microwave soldering process started with no tunning scenario in order to establish the resonant circuit manually after the calculation of the three stub tuners’ length, as is presented in Figure 8.

By monitoring the reflection of the incident microwave from samples back to the microwave generator, the lengths of stub tuners were calculated using Equation (5) [27]:(5)$$d_{i}=-\frac{\lambda }{2\pi }\cdot tan^{-1}\left(B_{i}\cdot Z_{0}\right)$$
where *d*_1_, *d*_2_, and *d*_3_ (mm) represent the length of the three stub tuners inside the WR340 waveguide; *λ* = 0.122 m is the wavelength of microwaves at 2450 MHz; *B*_1_, *B*_2_, and *B*_3_ (mS) are the susceptance of each stub having values between 0.1 mS to 1.0 mS depending on specific requirements of the application; and *Z*_0_ = 376.7 Ω is the air impedance. The computation of Equation (5) for insulating brick fire led to the following values: *d*_1_ = 0 mm, *d*_2_ = 19.83 mm, and *d*_3_ = 22.01 mm.

Using the values analytically determined by the lengths of the stub tuners, the microwave soldering process was started with microwave input power set to P_MW_ = 600 W, taking into consideration that the minimum level of injected microwave power was calculated to be set to P_MW_ = 598.89 W. The controlling unit of the microwave generator was set to stop automatically after reaching the limit soldering time of *t* = 60 s or limiting the soldering temperature to *T* = 240 °C. Then, for the rest sets of samples presented in Table 1, the microwave power was increased to 650 W, 700 W, and 750 W. Figure 9 presents a snapshot from the matching impedance tuning and the injected power which was recorded using Homer Software.

The temperatures recorded by the infrared pyrometer have revealed the stability of the indirect microwave soldering for all levels of power injected into the reaction chamber. However, for 750 MW, the heating process intended to suffer a thermal runaway of insulating fire brick. This phenomenon is well known and occurs during microwave heating of ceramics, leading to a faster increase in temperature and boosting the conversion rate of microwaves into heat [28,29,30]. Figure 10 presents the evolution of the temperature on the surface of copper plates during the joining process.

## 3. Results and Discussions

The discussions related to results obtained in the microwave soldering process of copper plates are focused on the quality of joints in terms of electrical resistance and tensile stress as electrical and mechanical properties obtained after soldering in the microwave field. In addition, an evaluation of economic costs related to the microwave soldering process was developed in order to be compared with the manual soldering process.

### 3.1. Electrical Resistance of Soldered Copper Plates 

The electrical resistance of the joined copper plates was measured following the same procedure presented in the previous paragraph. Taking into consideration that through overlapping the samples the total length of initial copper plates was reduced from 100 mm to 90 mm, the total electrical resistance without soldering should be lower. However, the contribution of a joint area, in terms of joint width and solder alloy, is significant for electrical resistance. The values measured for all samples are presented in Table 2.

The values of electrical resistance from Table 2 show that the total electrical resistance after the soldering process is similar to the initial electrical resistance of copper plates before cutting. The deviation of electrical resistance from the original value was calculated using Equation (6):(6)$$\varepsilon =\frac{\left|R_{i}-R_{w}\right|}{R_{i}}\cdot 100$$
where *ε* (%) is the deviation from the original electrical resistance, *R_i_* (Ω) is the initial electrical resistance of the copper plates before the cutting process, and *R_w_* (Ω) is the measured electrical resistance after microwave soldering. The results have shown that the deviation of electrical resistance was between 0 to 1% which is acceptable for the soldering process. The sample E4P2M presented a reduction of electrical resistance by 1.5% due to the fast heating and high power of the microwave. However, even the electronic applications require low-conductive joints, the microwave-injected power was too high and the process has become unstable. In addition, the width of the soldered joint was approximately double when compared with the average width of other soldered joints, which is not recommended in electronic applications due to the reduced space in integrated circuits. 

### 3.2. Evaluation of Tensile and Shear Characteristics of the Soldered Joints

The evaluation of the tensile and shear characteristics of the soldered joints was performed using test method standards for lead-free solders [31,32,33,34]. An LBG TC 100 electromechanical computerized universal testing machine from LBG srl (Azzano San Paolo, Italy, 2009) was used for the tensile testing. For the P1 samples heated at 600 W, the lower power acting on the ceramic support increases the heating time (Figure 10, curve E1P1M-E1P1T: the maximum temperature is reached in 55 s). This increase in heating time has a beneficial effect on the process: the soldering alloy, which is an alloy in close proximity to the eutectic alloy, melts suddenly and solidifies abruptly; if it remains in the molten state longer, it manages to absorb copper atoms by diffusion so that it moves slightly away from the chemical composition of the eutectic, resulting in slower cooling. This fact leads to a higher ductility of the joint material, which can be seen in the high value of the total elongation at fracture (Figure 11, curve E1P1M-E1P1T: the transverse stroke reaches 1.12 mm).

As the power injected into the ceramic substrate increases, a shift of the temperature peak to lower values of the time of reaching the maximum temperature is observed, together with a slight increase of the value of the maximum temperature. Shortening the heating time leads to a decrease in the ductility of the material, especially in samples E3P1 and E4P1, where powers of 700 W and 750 W were introduced, respectively. For these samples, the situation was very unpredictable, as the phenomenon of thermal runaway and even the formation of the microwave plasma occurred for the value of 750 W (Figure 11, curve E4P1M-E4P1T: sudden variations in the range of the maximum temperature marking the appearance of the microwave plasma). In addition, at the power of 700 W, there is a sudden drop in temperature in the joint, which reduces the ductility of the joint material (the total elongation at fracture drops to about half the value of the elongation measured in the sample produced at 600 W, from 1.12 mm to 0.6 mm). There are no significant differences between the values of total elongation at fracture for the 600 W and 650 W samples. Differences are noted in the values of maximum supported forces (0.82 kN for 600 W, compared to 0.60 kN for 650 W). However, it is interesting to note the maximum force value for the sample heated to 700 W, which is close to the value measured for the 600 W sample (0.77 kN compared to 0.82 kN), but three times higher than that for the test performed with a power of 750 W (0.77 kN compared to 0.25 kN).

The samples with a power of 650 W and 750 W show a drop in tensile force before reaching the maximum force, indicating either the previous presence of a discontinuity due to uneven or incomplete wetting or the occurrence of microcracks during the tensile test. This drop has no significant effect on the stress–strain curve, it returns to the initial value relatively quickly.

The pattern observed in the curves of the P1 specimens is also found in the curves of the P2 specimens. The only difference observed is the absence of microwave plasma, with all four curves showing clearly defined maxima.

The phenomenon of thermal runaway is observed in the samples produced with a power of 700 W and 750 W, respectively. For these samples, maximum values of 360 °C and 470 °C were reached during heating intervals of 26–27 s, respectively. In these situations, the soldering process is unpredictable, and the temperatures rise very quickly and uncontrollably.

At the 600 W and 650 W powers, the soldering process took place in a range between 40 and 55 s. As a result, high ductility was achieved for the sample heated at 600 W power (over 3 mm total elongation at break, according to Figure 12). However, there was a relatively large difference between the measured total elongation at fracture of the specimen heated with 600 W and the total elongation at fracture of the specimen heated with 650 W. In the second case, the total elongation at fracture was higher than in the first case. In the second case, the total elongation at break was about three times lower.

At the same time, the total elongation at break of the specimen produced with 750 W was about 10% of the total elongation at break of the specimen produced with 600 W, i.e., very large differences.

All samples show a drop in tensile strength before reaching maximum strength, indicating either the prior presence of a discontinuity due to uneven or incomplete wetting or the occurrence of microcracks during the tensile test.

Making a comparative analysis between the samples with solder alloy deposition on only one base material (P1) and those with deposition on both base materials (P2), the following was found:The general behavior during the mechanical tensile test of the two sets of samples is similar, an aspect confirmed by the appearances of the eight graphs (P1—4 graphs and P2—4 graphs).P1 samples show significantly lower maximum forces before breaking than P2 samples.The only noticeable difference is that in the case of the samples made with the power of 750 W where a P1 sample reaches the maximum force faster, but yields much more slowly, compared to a P2 sample which cracks suddenly, and the force drops to zero in 0.2 s.

The power of 600 W ensures proper heating of the soldering alloy by holding it for about 50–60 s. Additionally, it is necessary to ensure an intimate and firm contact between the two parts to be joined, but without applying excessive forces to avoid the expulsion of the liquid soldering alloy. After repeating the experiment for this type of sample for several thicknesses of the soldering alloy, it was found that the mechanical strength and ductility of the joint increase with the increase in the thickness of the deposited soldering alloy. Balancing this effect of the thickness of the deposited solder alloy layer with the measured values of the electrical resistance of the joint, it was found that a thickness in the range W_solder_ = (0.6 − 0.9) ∙ W_minBM_, where W_minBM_ represents the thickness of the thinnest base material, is a recommended value to harmonize the wetting process by obtaining a minimum electrical resistance. A differential melting of the soldering alloy deposited on the outside of the two base materials was observed compared to the melting of the alloy between the two base materials, most likely due to better heat transfer by conduction in the latter case. This differentiation produces fragmentation of the deposited solder alloy layer into two parts.

Although the layer of the soldering alloy may be of satisfactory thickness, its non-uniform deposition may reverse the wetting process, with the surface tension of the liquid metal between the solid copper surfaces drawing the alloy to a minimum volume instead of expanding it by wetting. Even if the heating is done at a temperature higher than the melting temperature of the soldering alloy, it is necessary to maintain this temperature for at least 20 s to ensure a correct wetting of the base material by the soldering alloy, especially in situations where the soldering alloy is deposited on a single base material.

Decreasing the thickness of the solder alloy layer deposited requires an increase in the pressing force to create firm contact between the two base materials and the solder alloy to ensure a better wetting of the surfaces of the base materials.

### 3.3. Macrographic Examination

The characterization of the soldered product was performed using macrographic examination [35,36]. The visual analysis of all the soldered samples led to the conclusion that a pattern related to material imperfections can be detected in a joint. This identified pattern allows the statement of conclusions applicable to categories of joints (P1 or P2, E1 or E2, or E3 or E4). Next, the discussions are presented on one example sample set E1/2/3/4P1/2. Table 3 shows examples of soldered joints from the two categories, P1 and P2, made with various strengths injected into the ceramic support.

In all P1 samples, the following characteristic aspects were identified, with greater or lesser intensity, mainly due to the power injected into the supporting ceramic:

The existence of areas with a lack of adhesion of the soldering alloy to the base material was found due to either a uniform deposit of the soldering alloy on the base material or due to improper preparation of the base material.

When the soldering alloy layer is uniformly deposited, proper wetting results; this wetting is not complete if the soldering alloy layer is insufficient. In such situations, certain discontinuities are identified in the mass of the soldering alloy.

In the case of applying excessive contact forces, it was found that ejections of the soldering alloy between the base materials occur. The ejected excess soldering alloy is usually bounded by the soldering alloy between the two base materials, indicating that the excess material was not formed during the soldering process, but was intentionally deposited prior to the soldering process. During the soldering process, reheating the soldering alloy upon microwave absorption produced the fragmentation of the previously deposited layer into two distinct parts. The shape of the excess material shows that the remelting of the soldering alloy was in a smaller volume at the level of the excess material compared to the alloy existing between the two base materials.

The following characteristic aspects were identified in all P2 samples. The existence of some areas with isolated or grouped pores was found due to the lack of protection of the bonding area; some pores can exceed 2 mm in diameter, for a solder that is only 2.2 mm thick, ending up covering a volume that stretches from one base material to the other. The wetting of the two surfaces of the base materials is no longer a problem, but when the layer of the soldering alloy is not uniformly deposited, the joint may have deviations from the usual shape. When excessive contact forces were applied, it was found that the solder alloy ejects between the base materials, as can be seen in the case of heating at 600 W.

A general visual examination of the fracture was performed as well. The fracture could reveal elements that influence the main mechanical characteristics of the joint. They can also provide important information regarding the allure of the stress–strain curve. By macrographic examination (examples of fracture areas are presented in Table 4), a lack of wetting was revealed for important areas of the P1 joints. That could be caused by two reasons: first, a too-short heating time during the soldering process (and that is specific E3 and E4 heat inputs), and second, an insufficient quantity of solder deposited on the copper plate. Both reasons could also be simultaneously happening. 

Poor wetting was detected for P2 samples as well, but the non-wet areas were low in dimension, and they are isolated or grouped. The rest of the flux was detected in several specimens. The presence of flux at the interface could produce an increase in the electrical resistance of the joint, which is not desired. All of these show the importance of a better preparation of the samples before the application of the soldering process.

### 3.4. Economical Sustainability of the Microwave Soldering Process

Microwave soldering technology is not cheap in terms of setting the soldering parameters in laboratory conditions, taking into consideration the high costs of microwave installation and auxiliary devices. However, from the point of view of the energy efficiency of the process, microwave soldering can be a sustainable process. Therefore, economic sustainability can be assessed by the calculation of energy consumption for the microwave soldering process.

The total energy consumed by microwave soldering can be calculated only for the process itself, taking into consideration that the preparation of samples, in terms of mechanical cutting and deposition of solder layers, is similar to manual soldering. Therefore, the total energy consumed can be assessed using Equation (7):(7)$$\overset{n}{\underset{i=1}{\sum }}P_{i}\cdot t=(P_{MW\;injected}+\overset{k}{\underset{j}{\sum }}P_{j})\cdot t$$
where *P_MW_*
_*injected*_ = 600 W represents the total injected microwave power in the soldering process; *P_j_* (W) represents the overall consumption of auxiliary devices such as infrared pyrometer (24 Vcc, 1 A), matching load impedance auto tuner (24 Vcc, 2.05 A), webcam and process computers (2 × 400 W including displays of computers); and *t* = 60 s represents the total time of soldering in the microwave field. The total energy consumed by the microwave soldering application will be:(8)$$E_{MWS}=\left(600\;W+24\;W+49.2\;W+800\;W\right)\cdot \frac{60}{3600}h=24.55\;\mathrm{Wh}$$

In order to assess the energy efficiency, the microwave soldering process is compared with a manual soldering process using the ProsKit 48 W/230 V digital soldering station, manufactured by ProsKit Industries (New Taipei City, Taiwan, 2016), with the temperature set at 371 °C. Manual soldering requires an average of 300 s soldering time and the total energy consumed is *E_MS_* = 4 Wh. Even manual soldering requires energy six times lower than microwave soldering, the sustainability of soldering in the microwave field is proved by the number of joints that can be obtained in one process. This can be done by designing a large heating chamber similar to domestic microwave ovens that can solder simultaneously with a high number of samples.

On the other hand, by establishing soldering parameters in the microwave field, the consumption of auxiliary devices can be reduced to 24 W, meaning the consumption of an IR pyrometer. The optimization of the microwave soldering process for one sample will require a total energy of *E_MWS_* = 10.4 Wh.

The economic sustainability of the microwave soldering process can be assessed considering 100 samples soldered simultaneously. Both processes, manual and microwave soldering, require one worker paid an average salary of 23 EUR/hour, in Europe [37].

Table 5 presents a comparative analysis of the total costs necessary for soldering copper plates. The cost of energy was also taken into consideration as the average of 0.25 EUR/kWh in the European Union [38].

## 4. Conclusions

Microwave soldering of copper plates is an advanced process that uses the interaction mechanism of microwaves with materials in order to develop heat for the joining process. Due to the high reflection of microwaves by metal materials, microwave soldering is an indirect process, with the heat for joining being developed by ceramic susceptors.

The results of temperature simulation for 60 s, thermal transient, and temperature distribution revealed a temperature of 731 °C in the center of the fire brick block and 240 °C on the surface of copper plates. Using findings from the simulation model, the level of microwave power was determined to be at least 598.89 W. The experimental program has consisted of microwave heating of assembly fire brick–copper plates at 600 W, 650 W, 700 W, and 750 W for a soldering time limited to 60 s.

The results obtained have indicated a small variation up to ±1.5% of the electrical resistance of the joints compared with the initial values before soldering. These results led to the conclusion that microwave soldering does not significantly influence the total electrical resistance, which is mandatory for electronic applications.

Making a comparative analysis between the samples with solder alloy deposition on only one base material (P1) and those with deposition on both base materials (P2), the results have shown that the general behavior during the mechanical tensile test of the two sets of samples is similar. The P1 samples presented significantly lower maximum forces before breaking than the P2 samples; the only noticeable difference is in the case of the samples made with the power of 750 W where the P1 samples reaches the maximum force faster, but yields much more slowly, compared to the P2 sample which cracks suddenly and the force drops to zero in 0.2 s.

Finally, microwave soldering was evaluated from an economic point of view. The economic sustainability of the process, in terms of wide application in the electronic industry, was assessed through the calculation of the costs related to electricity consumption and the salary paid for the process. A total cost of 0.3859 euro/100 joints was calculated, which is cost-effective compared with manual soldering. Additionally, microwave soldering can be fully automated/robotized, which creates an advantage from the digital manufacturing point of view.

## Figures and Tables

**Figure 1 materials-16-03311-f001:**
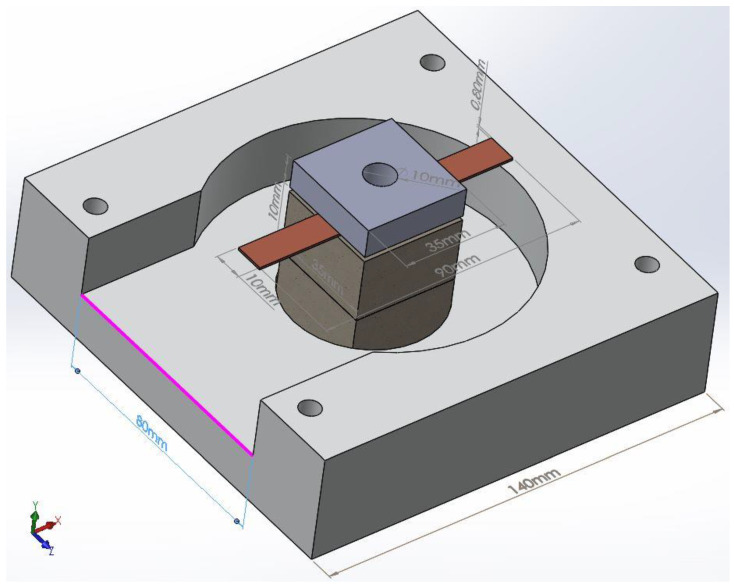
The model developed for temperature distribution and thermal transient simulation during microwave soldering (overall dimensions of heating chamber 140 mm × 140 mm × 90 mm).

**Figure 2 materials-16-03311-f002:**
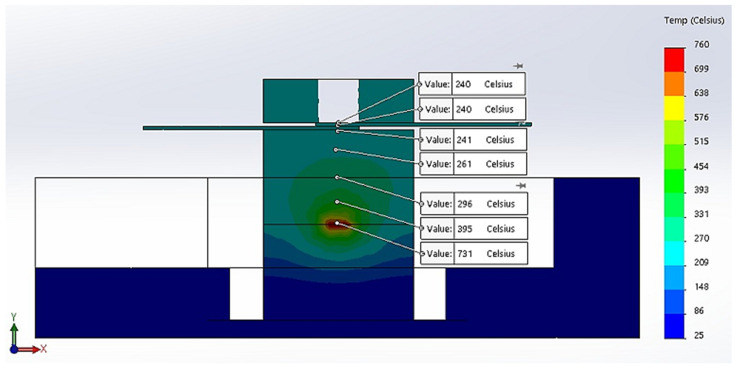
Thermal transient and temperature distribution for insulating brick fire material.

**Figure 3 materials-16-03311-f003:**
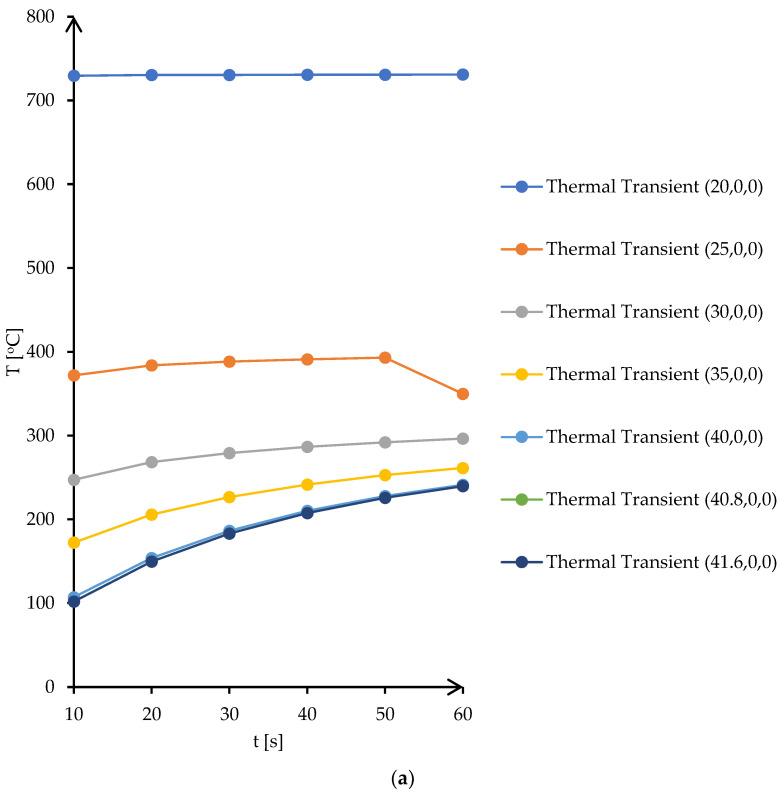
Transient temperature and temperature distribution in insulating brick fire material: (**a**) transient temperature of vertical *Y*-axis points for 10 s step time, (**b**) temperature distribution on *Y*-axis locations after 60 s.

**Figure 4 materials-16-03311-f004:**
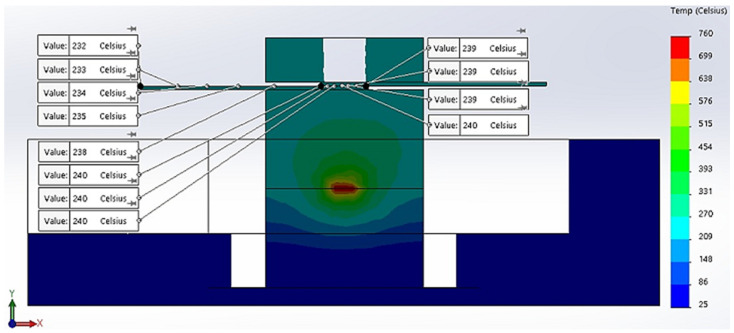
Thermal transient and temperature distribution for copper plates.

**Figure 5 materials-16-03311-f005:**
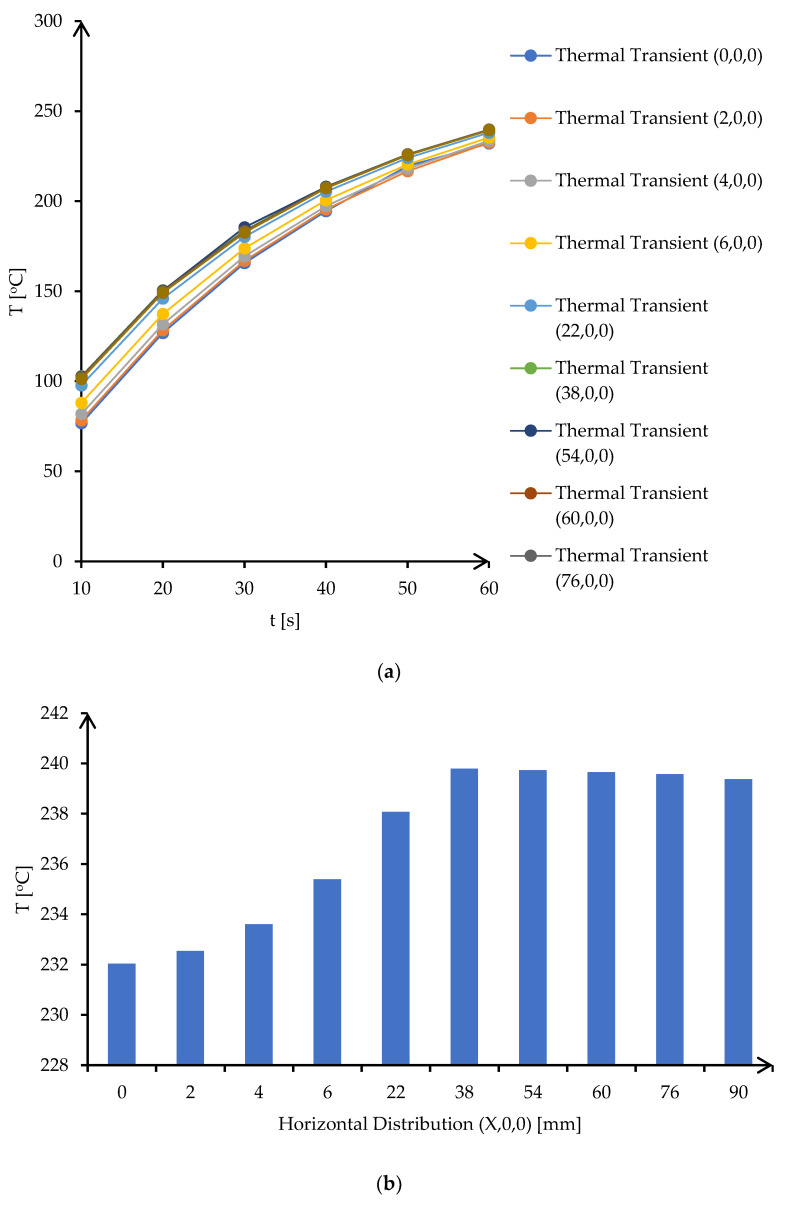
Transient temperature and temperature distribution in copper plates: (**a**) transient temperature of the horizontal *X*-axis points for 10 s step time, (**b**) temperature distribution on the *X*-axis locations after 60 s.

**Figure 6 materials-16-03311-f006:**
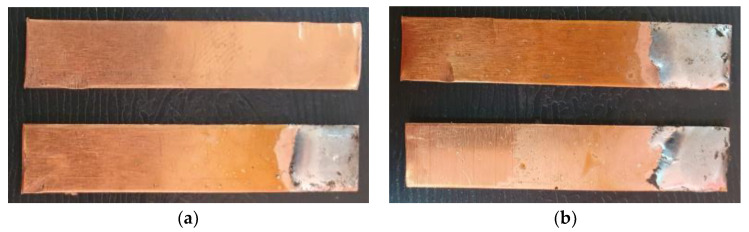
Copper plates samples prepared for microwave soldering process: (**a**) SAC305 layer deposited on one sample (code sample EP1M and EP1T), (**b**) SAC305 layer deposited on both samples (code sample EP2M and EP2T).

**Figure 7 materials-16-03311-f007:**
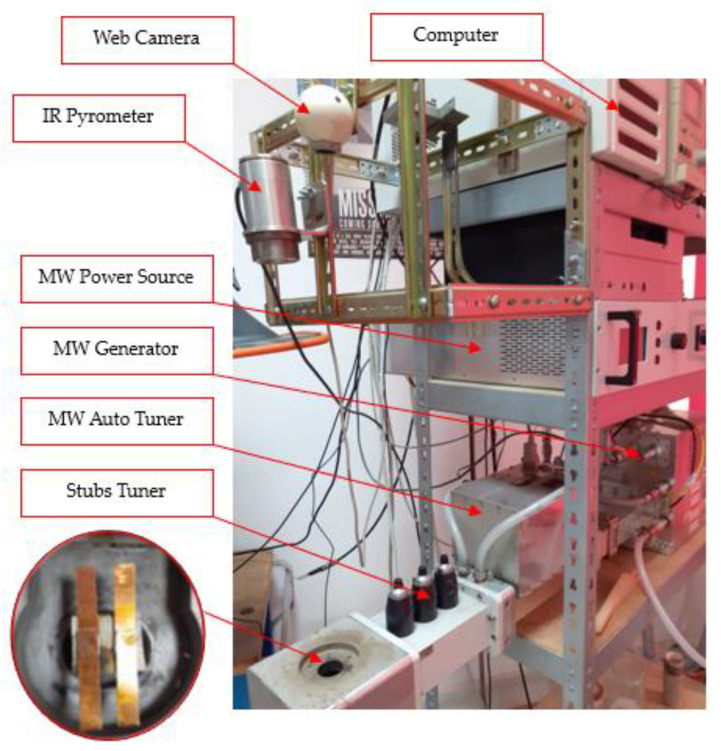
Experimental microwave soldering installation.

**Figure 8 materials-16-03311-f008:**
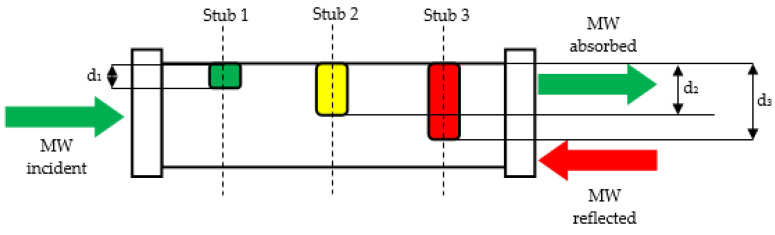
Matching load impedance using three stub tuners.

**Figure 9 materials-16-03311-f009:**
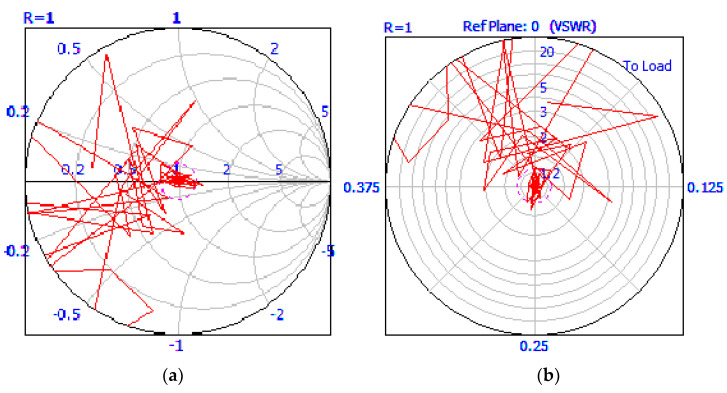
Microwave power required by the soldering process: (**a**) power transferred to samples, (**b**) power absorbed and converted into heat by insulating fire brick, (**c**) balance powers of the microwave soldering process.

**Figure 10 materials-16-03311-f010:**
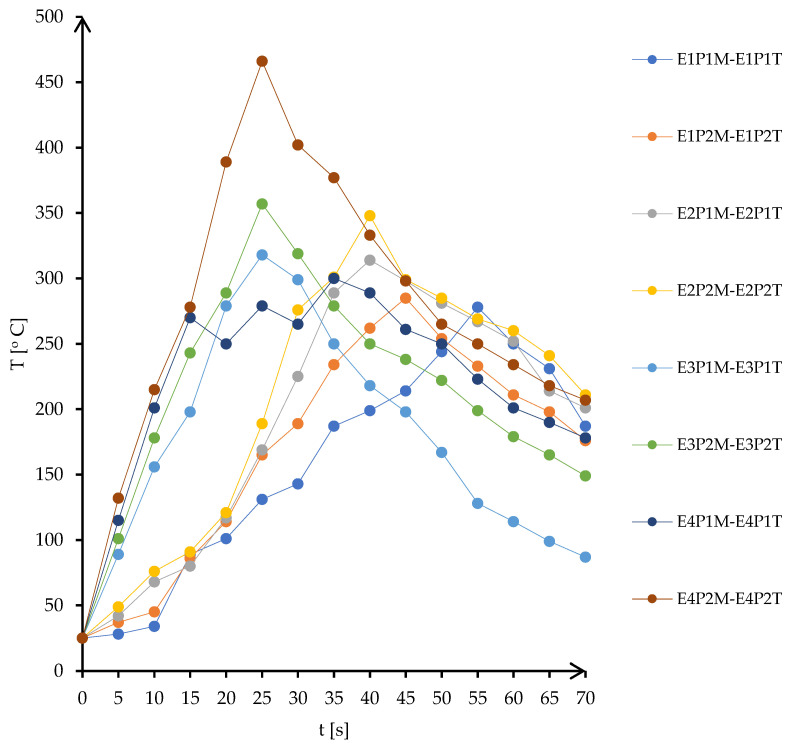
Temperatures measured by IR pyrometer on the surface of joined copper plates.

**Figure 11 materials-16-03311-f011:**
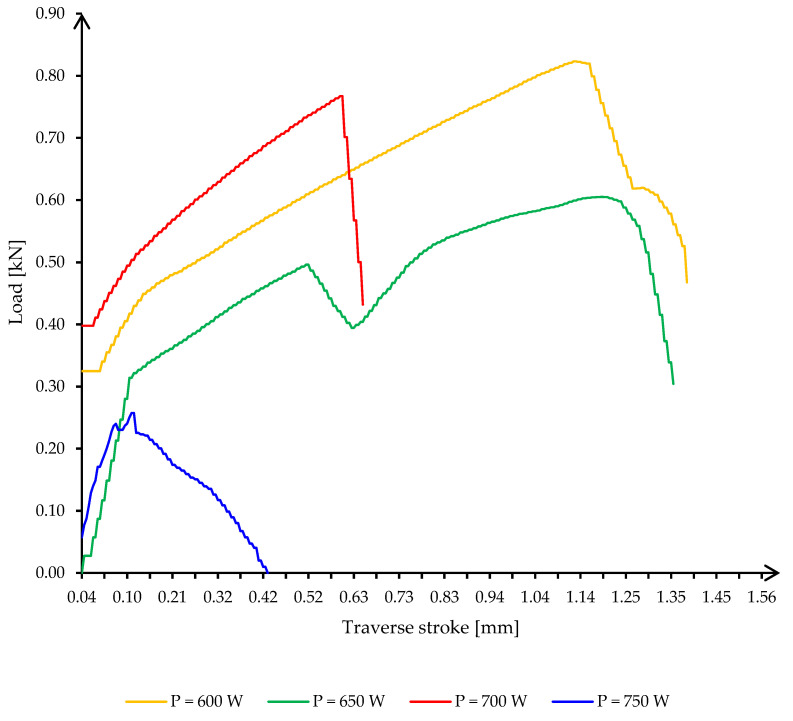
Curves load against the transverse stroke of the P1 specimens, for a solder layer of 1.0 mm thickness.

**Figure 12 materials-16-03311-f012:**
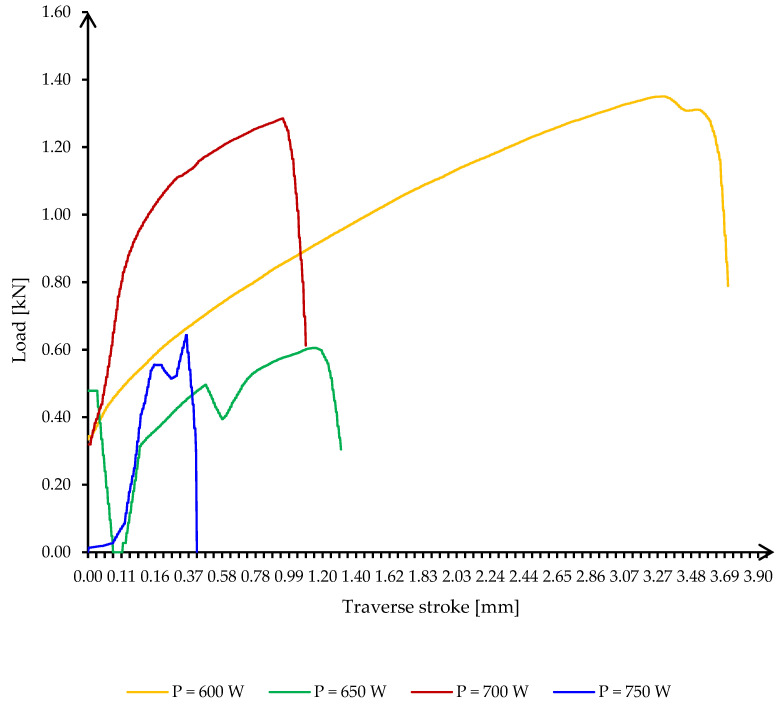
Curve loads against the transverse stroke of the P2 specimens, for a solder layer of 2.2 mm thickness.

**Table 1 materials-16-03311-t001:** Initial parameters of samples used in the indirect microwave soldering process.

Code	R_i_ (Ω)	W_i_ (mm)	R_iS1_ (Ω)	W_S1+SAC305_ (mm)	R_iS2_ [Ω]	W_S2+SAC305_ [mm]
P_MW_ = 600 W						
E1P1T	0.0400	0.8	0.0200	1.68	0.0200	0.8
E1P1M	0.0400	0.8	0.0200	1.73	0.0200	0.8
E1P2T	0.0400	0.8	0.0200	1.50	0.0200	1.59
E1P2M	0.0400	0.8	0.0200	1.71	0.0200	1.69
P_MW_ = 650 W						
E2P1T	0.0400	0.8	0.0200	1.53	0.0200	0.8
E2P1M	0.0400	0.8	0.0200	1.67	0.0200	0.8
E2P2T	0.0400	0.8	0.0200	1.51	0.0200	1.55
E2P2M	0.0400	0.8	0.0200	1.65	0.0200	1.65
P_MW_ = 700 W						
E3P1T	0.0400	0.8	0.0200	1.41	0.0200	0.8
E3P1M	0.0400	0.8	0.0200	1.40	0.0200	0.8
E3P2T	0.0400	0.8	0.0200	1.33	0.0200	1.34
E3P2M	0.0400	0.8	0.0200	1.44	0.0200	1.25
P_MW_ = 750 W						
E4P1T	0.0400	0.8	0.0200	1.45	0.0200	0.8
E4P1M	0.0400	0.8	0.0200	1.27	0.0200	0.8
E4P2T	0.0400	0.8	0.0200	1.43	0.0200	1.46
E4P2M	0.0400	0.8	0.0200	1.47	0.0200	1.43

**Table 2 materials-16-03311-t002:** The electrical resistance of the joined samples measured using the “two points” method.

Code	R_i_ (Ω)	R_w_ (Ω)	W_w_ (mm)	ε (%)
E1P1T	0.0400	0.0398	1.89	0.5
E1P1M	0.0400	0.0403	2.11	0.75
E1P2T	0.0400	0.0396	1.81	1
E1P2M	0.0400	0.0397	1.81	0.75
E2P1T	0.0400	0.0400	2.59	0
E2P1M	0.0400	0.0402	1.84	0.5
E2P2T	0.0400	0.0404	2.74	1
E2P2M	0.0400	0.0401	1.74	0.25
E3P1T	0.0400	0.0398	1.72	0.5
E3P1M	0.0400	0.0400	1.64	0
E3P2T	0.0400	0.0396	2.22	1
E3P2M	0.0400	0.0397	2.25	0.75
E4P1T	0.0400	0.0397	1.90	0.75
E4P1M	0.0400	0.0397	1.86	0.75
E4P2T	0.0400	0.0397	1.80	0.75
E4P2M	0.0400	0.0394	3.67	1.5

**Table 3 materials-16-03311-t003:** Examples of soldered joints containing material imperfections.

MW Power (W)	One Side Deposited Solder (P1)	Double-Side Deposited Solder (P2)
600	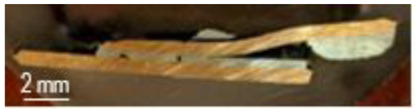	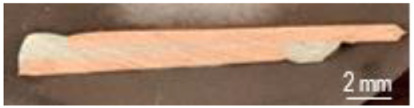
650	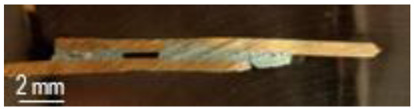	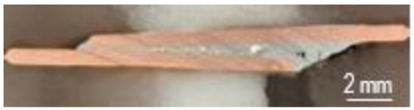
700	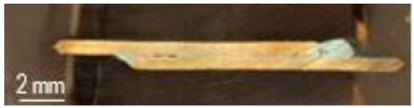	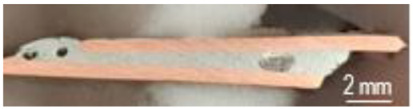
750	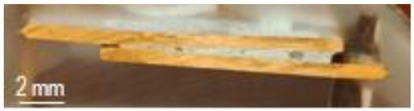	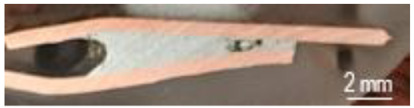

**Table 4 materials-16-03311-t004:** Examples of fractures for P1 and P2 tensile tested specimens.

One Side Deposited Solder (P1)	Double-Side Deposited Solder (P2)
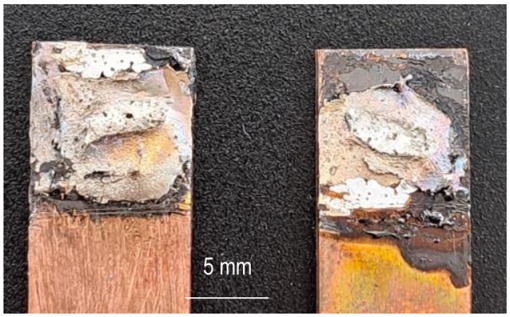	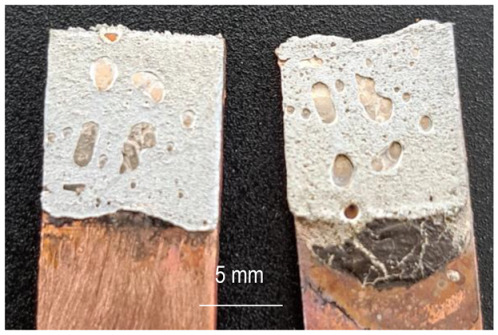
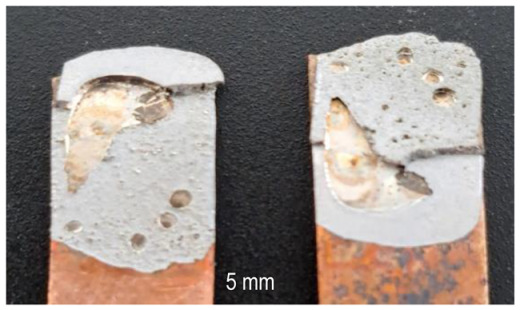	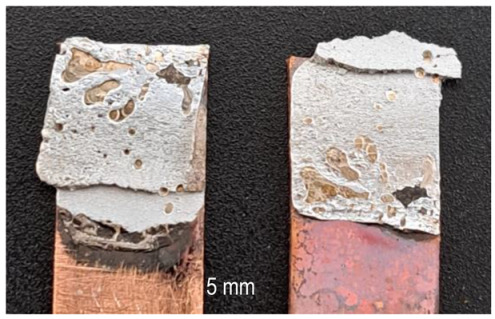

**Table 5 materials-16-03311-t005:** Total costs required by manual and microwave soldering.

Soldering Process	Energy (EUR)	Personnel (EUR)	Total (EUR)
MS (1 sample)	0.0010	1.9166	1.9176
MWS (100 samples)	0.0026	0.3833	0.3859

## Data Availability

Not applicable.

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
