# Peer review of "Microwave Soldering of Low-Resistance Conductive Joints—Technical and Economic Aspects"

_materials, 2023, doi:10.3390/ma16093311_

Round 1
Reviewer 1 Report
Authors have presented an article entitled “Microwave soldering of low-resistance conductive joints-technical and economic aspects” Though the manuscript is well written and organized but there is scope for further improving the quality of the draft before considering for publication.
Few minor comments are listed below:
1. As author mentioned that ceramic materials having potential applications in microwave absorption and sensors, polymeric materials also having same features. Author should send the performance of these articles to make it contrast: Composites Part A: Applied Science and Manufacturing, 107427 (2023); Polymer Bulletin 76, 3621-3642 (2019).
2. In Figure 1, author described the model of microwave soldering, and there are few measurements, such as 140, 80, etc. The units should be specified along with the measurements.
3. There are six different curves in Figure 3a; please mention the caption in graphs like Figure 10.
4. In both cases P1 and P2, the load value is less in 750W, but in 600W having high elongation. Hence, increasing power affects the toughness of solder. Please explain in detail.
5. The conclusion is very lengthy, try to concise it with using bullets point in the revised version.
6. Authors should check the typos and grammatical mistakes.
Minor correction required for typos and grammatical mistakes.
Author Response
Dear Reviewer
Thank you for valuable comments and suggestions. Please find attached the answers to all remarks.
Authors

Reviewer 2 Report
Dear authors,
the presented article is interesting from the view of utilisation of microwave energy for joining creations.
I have questions:
1.Can you define the chemical composition ( material properties) of tested copper samples and soldered layers?
2. Did you clean the testing surfaces before the soldering on copper samples or not? It can create the inpurities, bubles in the soldered layers or imperfectly connectionsi n the joins.
3. Was the thickness of the soldered layer coating ( 0,7 mm) the same on the applied copper sample 10x10 mm , or was the soldered thickness in the center greater and at the edges smaller or missing? How did you achieve the application of the same applied solderd thickness?
4.Which type of microwave machine/apparatus did you use for experiments?
Author Response

(The authors gave the same response as above.)
